



# Novel Statistical Analysis Illustrates Importance of Flow Source for Extreme Variation in Dissolved Organic Carbon in a Eutrophic Reservoir in the Great Plains

Anthony A. P. Baron[1], Helen M. Baulch[1,2], Ali Nazemi[3], Colin J. Whitfield[1,2]

[1]School of Environment and Sustainability, University of Saskatchewan, Saskatoon, S7N 5C5, Canada.
[2]Global Institute for Water Security, University of Saskatchewan, Saskatoon, S7N 3H5, Canada.
[3]Department of Building, Civil and Environmental Engineering, Concordia University, Montréal, H3H 2L9, Canada.

*Correspondence to*: Anthony A. P. Baron (anthony.baron@usask.ca)

**Abstract.** Long-term dissolved organic carbon (DOC) trends have been observed across many regions of the Northern Hemisphere, yet the drivers of these trends are not universal. Elevated DOC concentrations are a major concern for drinking water treatment plants that draw from freshwaters, owing to effects on disinfection byproduct formation, risks of bacterial regrowth in water distribution systems, and increasing treatment costs. Using a unique 30-year data set encompassing both extreme wet and dry conditions in a eutrophic drinking water reservoir in the Great Plains of North America, we investigate the effects of changing source water and in-lake water chemistry on DOC. Using wavelet coherence analyses and generalized additive models of DOC, we find DOC concentration was significantly coherent with flow from a large upstream mesotrophic reservoir. DOC was also coherent with sulfate, total phosphorus, ammonium, and chlorophyll *a* concentrations across the 30-year record. These variables accounted for 56% of the deviance in DOC from 1990 to 2019, suggesting that water source and in-lake nutrient and solute chemistry are effective predictors of DOC concentration. Clearly, climate and changes in water and catchment management will influence source water quality in this already water-scarce region. Our results highlight the importance of flow management to shallow eutrophic reservoirs. They also highlight a key challenge where wet periods can exacerbate water quality issues and these effects can be compounded by reducing inflows from systems with lower DOC. These flow management decisions address water level and flood risk concerns but have important impacts on drinking water treatability.



## 1 Introduction

Numerous studies over the last few decades have assessed long-term trends in dissolved organic carbon (DOC) in freshwater lakes and reservoirs in the Northern Hemisphere, primarily in forested regions. Several studies report increasing DOC concentrations across regions including eastern Canada (Couture et al., 2011; Hudson et al., 2003), the northeastern United States (Rodríguez-Cardona et al., 2022), northern Europe (Futter et al., 2014; Pärn and Mander, 2012), central Europe (Hruška et al., 2009), and the United Kingdom (Evans et al., 2005). Others report no significant trends in DOC concentration (Dillon

and Molot, 2005; Jane et al., 2017), that trends in flux don't accompany trends in concentration (Eimers et al. 2008), or report decreasing trends over time (Rodríguez-Murillo et al., 2015). These different findings across systems have sparked much debate over the factors that govern DOC concentrations, and have led to a search for common drivers (e.g., Pagano et al., 2014; Temnerud et al., 2014; Winterdahl et al., 2014). One driver receiving substantial focus has been declining anthropogenic sulfur dioxide emissions and consequent declines in deposition (Monteith et al., 2007). Trends in DOC have also been linked to

climatic and hydrological factors such as changes in temperature, precipitation, or runoff (Hongve et al., 2004; Weyhenmeyer and Karlsson, 2009), cycles in sea salt (i.e., chloride) deposition (Evans et al., 2006), atmospheric nitrogen (N) deposition (Evans et al., 2008), land management and land use (Yallop and Clutterbuck, 2009), and increased atmospheric carbon dioxide concentrations (Freeman et al., 2004).

Aquatic DOC export is highly correlated with precipitation and annual runoff, hence is regulated by local hydrology and

climate, as well as landscape features which influence DOC production and transport (Pace and Cole, 2002; Porcal et al., 2009). At the landscape scale, allochthonous DOC leached from terrestrial soils is transported to streams and lakes, and DOC flux from soils is regulated by soil moisture and flow paths from soil to stream or lake (Clark et al., 2010; Sobek et al., 2007). Landscape features that influence DOC concentration in lakes include drainage ratio (catchment:lake area), hydrological connectivity of water sources such as wetlands and streams (Laudon et al., 2011; Schiff et al., 1997), land-use (Williams et al.,

2010; Wilson and Xenopoulos, 2009), wetland area (Dillon and Molot, 1997), proportion of the catchment that is open water (Kortelainen, 1993), and watershed slope and topography (Sobek et al., 2007; Xenopoulos et al., 2003).

This growing body of research suggests that landscape complexity can contribute to complexity in DOC exports. The Canadian prairie region, a vast area of $1.8 \times 10^6$ km$^2$, features flat topography, poorly developed stream networks, abundant depressional storage in pothole wetlands and climatic variability. Periods of deluge and drought are common (Pham et al., 2009; Pomeroy

et al., 2007; Vogt et al., 2018) and soils of the region often have high concentrations of DOC and $SO_4^{2-}$, which is reflected in pothole pond chemistry (Arts et al., 2000; Labaugh et al., 1987; Waiser, 2006). Importantly, approximately half of the region is hydrologically non-effective (does not contribute flow to larger streams in a typical year), hence surface flows from the local catchment are irregular and typically limited to a few weeks in spring. As such they may play an important, albeit irregular, role in delivering nutrients and DOC to larger water bodies. Although endorheic saline prairie lakes are known for their often

extremely high DOC concentrations owing to prolonged evapoconcentration (Arts et al. 2000; Osburn et al. 2011; Waiser and

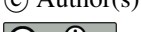



Robarts 2000), DOC concentrations in freshwater lakes of the region have received less attention, and few long time series of DOC concentration exist. Long-term change in DOC is of particular interest to drinking water treatment providers relying on surface water in the region. Understanding these DOC patterns can support safe drinking water treatment and limit production of disinfection byproducts like trihalomethanes and haloacetic acids. Given dissolved organic matter (DOM) is a master

variable in aquatic systems (e.g., Prairie, 2008; Wetzel, 2001a) and given ongoing changes in climate, future changes in DOC are also of interest owing to the potential to affect limnological behaviour of lakes in the region.

This study investigates long-term variation in DOC concentration in Buffalo Pound Lake, (Saskatchewan, Canada), a lake with high DOC for a drinking water source, and a lake that is prone to cyanobacterial blooms (Painter et al. 2022a). Our objectives were to understand relationships between DOC, flows from upstream Lake Diefenbaker and the local catchment, and a suite

of in-lake water chemistry parameters over a 30-year period (1990–2019) encompassing both wet conditions and severe drought, with varied water flows and management. Our analysis centres on the novel use of wavelet coherence and phase analyses to identify temporally-dependent patterns between time series that are typically masked by traditional correlation methods. These relationships were further investigated with generalized additive modeling (GAM) to understand the predictor-response relationship between DOC concentration, flows, and water chemistry. We hypothesized that at longer timescales

flows into Buffalo Pound Lake would be important determinants of patterns in DOC concentration, while changes in lake water chemistry would impact DOC at shorter timescales.

## 2 Methods

### 2.1 Study site

Buffalo Pound Lake is a shallow eutrophic reservoir located near the headwaters of the Qu'Appelle River drainage system in

southern Saskatchewan, Canada (50.65127º N, 105.508225º W) (Fig. 1). Buffalo Pound Lake is the drinking water source for ~25% of the province's residents, servicing the cities of Regina (population 249,000) and Moose Jaw (population 34,000). The lake is the site of a provincial park and a popular recreation spot for activities that include fishing, swimming, and camping. Climate in this region is subhumid continental with long, cold winters and short, warm summers. Mean annual precipitation is 320 mm (McGowan et al., 2005), but varies between dry and wet climate phases, and average annual temperature is ~3ºC

(Haig et al., 2021). Buffalo Pound Lake was created by damming a natural lake's outflow to the Moose Jaw River to raise water levels beginning in 1939. The lake is long and narrow (~29 km by 1 km) with an average depth of 3 m (max depth 5.6 m) and storage capacity of $9\times10^7$ m³ (BPWTP, 2020). Its shallow depth and exposure to regular wind-induced mixing result in a polymictic system that only periodically establishes thermal stratification (Finlay et al., 2019). Buffalo Pound Lake drains a 3310 km² agricultural catchment with nutrient-rich soils that contribute to nutrient influx and eutrophic conditions (Finlay et

al., 2019; Hammer, 1971). The nature of the catchment's contribution to runoff generation is extremely complex; however, and the lake's effective catchment area, defined as that portion of the watershed that contributes flow in 1:2 runoff years (Godwin and Martin, 1975), is just 38% of the gross catchment area.





Flows at Buffalo Pound Lake are managed to maintain lake levels within an established operating range (30 cm) that has some seasonal variability. During the summer months, due to the importance of the water supply and recreational interests, levels are typically managed within the upper 10 cm of the operating range, but are reduced by about 15–20 cm in winter. The dominant water source to the lake used to control lake levels is releases from the Qu'Appelle River Dam on Lake Diefenbaker reservoir (Fig. 1), lying outside of the natural catchment boundary (BPWTP, 2020). Between 2015–2020, mean annual water release from Lake Diefenbaker ranged from 1.8–4.8 m$^3$ s$^{-1}$ (BPWTP, 2021). The lake also experiences transient inflows from the local catchment during spring snowmelt and (infrequent) extreme summer precipitation. During wet years, with high catchment flows to Buffalo Pound, managed releases from Lake Diefenbaker are much lower than in dry years. Dry years, with little or no catchment flow, require larger releases from this upstream reservoir to maintain lake levels within the operating range. Water quality characteristics of the local catchment, including DOC concentration and DOM quality, are distinct from those of Lake Diefenbaker.





**Figure 1. Map of Buffalo Pound Lake, Saskatchewan, Canada, highlighting its gross (grey) and effective (green) drainage areas, including upstream Lake Diefenbaker and the Qu'Appelle River. The middle of Buffalo Pound Lake is located at 50.65127º N, 105.508225º W. Red diamonds denote Water Survey of Canada (WSC) streamflow gauging stations in the Buffalo Pound Lake watershed with complete or reconstructed records over the study period (see Sect. 2.3 and Table S1). The inset map of Canada was plotted using the R package *mapcan* (McCormack and Erlich, 2019). Gross and effective drainage area shapefiles were obtained from the Prairie Farm Rehabilitation Administration (PFRA). Lake and river geospatial data were retrieved through the Government of Canada's Open Government License (Statistics Canada, 2006, 2011).**



## 2.2 Data

### 2.2.1 Water chemistry

Water samples were collected and analyzed on weekly, semi-monthly, or monthly intervals at the Buffalo Pound Water
Treatment Plant (BPWTP) between 1990 and 2019. The BPWTP laboratory is accredited to the ISO/IEC 17025 standard and
adapts their analysis methods using Standard Methods for the Examination of Water and Wastewater (APHA, 2012). The
accredited BPWTP laboratory did change instruments over the long history of this data set; however, new instruments were
cross validated and subjected to testing requirements associated with accreditation (B. Kardash pers. comm.). We describe
briefly here the current analytical methods (Table S2). Prior to 2015, DOC samples were analyzed via nondispersive infrared
detector on a Tekmar/Dohrmann Phoenix 8000 Carbon Analyzer. Since 2015, DOC concentrations have been analyzed on a
General Electric (GE) M5310M Laboratory total organic carbon (TOC) Analyzer with GE-patented Sievers Selective
Membrane Conductometric design. Sulfate ($SO_4^{2-}$) and nitrate ($NO_3^-$) concentrations (as S and N respectively) were
determined via ion chromatography using a Dionex ICS-1100 Ion Chromatograph. Ammonium ($NH_4^+$ as N) concentrations
were determined colorimetrically by Nessler's reagent addition after samples are buffered in boric acid solution and distilled.
Chlorophyll $a$ (Chl $a$) pigment concentrations were extracted with acetone and measured via spectrophotometry, but were not
corrected for the presence of pheophytin. Total phosphorus (TP) analysis was via molybdenum blue method at 690 nm on a
spectrophotometer after digestion. Soluble reactive phosphorus (SRP) concentrations were analyzed using the ammonium
molybdate method then measured at 690 nm on a spectrophotometer.

The frequency of water chemistry analyses permitted using monthly averages ($n = 360$) across the 30-year time series with
few missing data points (0.28–3.6%) for the seven parameters described above. Statistical analyses (described below) required
a complete record, necessitating (very limited) imputation. Because many water chemistry parameters were highly variable
over decadal scales, replacing missing values with (seasonal) mean or median consistently produced values outside the
reasonable range for a given month and year based on visual inspection. To overcome this the $k$-nearest neighbour regression
approach (Altman, 1992; Fix and Hodges, 1951) was used, replacing missing values with the mean of $k = 1$ nearest neighbours.

### 130 2.2.2 Streamflow measurements

The streamflow monitoring network in the Buffalo Pound Lake catchment is operated by the Saskatchewan Water Security
Agency (WSA), and comprises four streamflow gauges with sometimes incomplete records for 1972–2019 (Fig. 1, Table S1).
Lake Diefenbaker outflow ($Q_{LD}$) and Buffalo Pound Lake inflow ($Q_{BP}$) are gauges at the upstream and downstream reaches of
the Qu'Appelle River that represent inflow from Lake Diefenbaker, and inflow to Buffalo Pound Lake, respectively. Flows
from Ridge Creek ($Q_{RC}$) and Iskwao Creek ($Q_{IC}$) are tributaries sourced from the local catchment that flow into the Qu'Appelle
River between Lake Diefenbaker and Buffalo Pound Lake (see Sect. 2.3).



## 2.3 Streamflow reconstruction scheme

Conventional hydrological modeling is not an easy task in the Buffalo Pound Lake catchment due to the poorly-defined drainage system distributed over a relatively flat terrain (e.g., Mekonnen et al., 2015; Pomeroy et al., 2005). This forms a complicated hydrological condition, in which the hydrologic responses heavily depend on landscape feature detail (Fang et al., 2007; van der Kamp et al., 2003; Spence, 2010), and are subject to nonlinear hysteresis (Shook and Pomeroy, 2011). With few exceptions (Clark and Shook, 2022), current hydrological models do not represent dominant hydrological processes in prairie landscapes and fail in reproducing the observed runoff. Here we attempted to maximize the use of observed gauge information (Table S1) by implementing a data-driven reconstruction scheme. We used available flow information at Ridge Creek ($Q_{RC}$) to estimate missing flows at Iskwao Creek ($Q_{IC}^*$). Likewise, we used Ridge Creek to estimate flows for the ungauged portion of the catchment upstream of the Buffalo Pound Lake inflow (($Q_U - \frac{\Delta S}{\Delta t})^*$), an area that includes Eyebrow Lake (Fig. 1). Considering the mass conservation and ignoring other potential losses, the upstream inflow to Buffalo Pound Lake ($Q_{BP}$) can be described as:

$$Q_{BP} = Q_{LD} + Q_{RC} + Q_{IC} + Q_U - \frac{\Delta S}{\Delta t}, \tag{1}$$

where the storage term ($\frac{\Delta S}{\Delta t}$) accounts for changes in landscape storage, including at Eyebrow Lake. While records for $Q_{LD}$ and $Q_{RC}$ are complete, we used a set of linear and nonlinear transfer functions, including polynomial, exponential, inverse exponential, along with power and tangential functions and their linear combinations (total of 420 functional mappings) to link $Q_{IC}^*$ and $(Q_U - \frac{\Delta S}{\Delta t})^*$ to $Q_{RC}$. We note that the nature of landscape hydrologic response in the lake catchment is significantly different between frozen (December to March) and ice-free (April to November) months and therefore, we considered separate transfer functions for warm and cold seasons. The 1972–1992 data were used for model development and the models compared and falsified based on Bayesian Information Criterion that penalizes the number of model parameters. For Iskwao Creek, the following transfer functions stood up during the training period:

$$Q_{IC}^* = \begin{cases} 0.3112 Q_{RC}^{0.4537} & \text{if cold season} \\ 0.4957 Q_{RC} + 0.1185 & \text{if warm season and } Q_{RC} > 0 \\ 0 & \text{if warm season and } Q_{RC} = 0 \end{cases}, \tag{2}$$

where $Q_{IC}^*$ and $Q_{RC}$ are in $m^3 \, s^{-1}$. Similarly, the total ungauged flux ($m^3 \, s^{-1}$) was estimated based on $Q_{RC}$ as:

$$(Q_U - \frac{\Delta S}{\Delta t})^* = \begin{cases} 0.01292 Q_{RC}^3 - 0.303 Q_{RC}^2 + 1.249 Q_{RC} - 0.0956 & \text{if cold season} \\ -0.05958 Q_{RC}^2 + 2.77 Q_{RC} - 0.1463 & \text{if warm season} \end{cases}, \tag{3}$$

The efficiency of estimated $Q_{IC}^*$ and $(Q_U - \frac{\Delta S}{\Delta t})^*$ was investigated through calculating the Coefficient of Determination ($R^2$) for observed flows at Iskwao Creek and upstream of Buffalo Pound Lake during separate training (1972–1992) and testing periods. $R^2$ measures the percentage of described variance, being 0.71 during the training and 0.68 during the testing period of 2008 to



2011 for $Q_{IC}^*$. For the total inflow to the lake ($Q_{BP}$), $R^2$ was similarly high, 0.87 during training and 0.78 during testing (1993–
1995).

Finally, because $Q_{BP}$ is dominated by $Q_{LD}$, we used a third metric as a means of assessing the role of flows from the local
catchment. In this instance we assume that $Q_{RC}$ and $Q_{IC}$ are representative of the wider catchment, and can be scaled and used
to approximate the local catchment flows ($Q_{LC}$), including runoff from the catchment area downstream of the gauge at the
Buffalo Pound Inflow ($Q_{BP}$):

$$Q_{LC} \cong Q_{RC} + Q_{IC} + Q_{BP}, \tag{4}$$

Here, we scale combined flows from Ridge and Iskwao Creeks according to effective catchment areas to estimate $Q_{BP}$,
including effective areas along either side of the lake. While this approach does not offer precise timing of these flows reaching
Buffalo Pound Lake due to potential for some attenuation in upstream reaches on the Qu'Appelle River, it nonetheless provides
a proxy for flow contribution from the local catchment.

## 2.4 Statistical analyses

There were two major goals of our statistical analysis. First, we aimed to understand frequency-dependent relationships
between DOC concentration and a suite of environmental predictors that include upstream flows ($Q_{LD}$, $Q_{BP}$, $Q_{LC}$) and in-lake
water chemistry ($SO_4^{2-}$, TP, SRP, Chl $a$, $NO_3^-$, $NH_4^+$) using wavelet coherence analysis (Grinsted et al., 2004; Sheppard et al.,
2016, 2017; Walter et al., 2021). Wavelet-based methods can be used to measure synchrony and coherence between
environmental and biological variables, and investigate relationships that are not readily detected by conventional correlation
methods (Reuman et al., 2021). Specifically, wavelet coherence analysis can reveal the direction (analogous to positive or
negative correlation) and phase (positive or negative time-lag) between two variables at different timescales. Predictors
identified as significantly coherent with DOC ($\alpha = 0.05$) by wavelet coherence analysis were subsequently used toward our
second objective wherein generalized additive modeling (GAM) was used to investigate predictor–response relationships
between DOC concentration, flows, and in-lake water chemistry.

All statistical analyses were carried out in R version 4.1.3 (R Core Team, 2022), including Wavelet Approaches to Synchrony
(*wsyn*) for wavelet analysis (version 1.0.4; Reuman et al., 2021), *base* (R Core Team, 2022), and *spatstat.core* (version 2.4.2;
Baddeley et al., 2015) for wavelet plotting. Generalized additive models were fit using the R package *mgcv* (version 1.8.40;
Wood, 2017), and graphics were plotted with *ggplot2* (version 3.3.6, Wickham, 2016), *gratia* (version 0.7.3; Simpson, 2021),
and *patchwork* (version 1.1.1; Pedersen, 2020). Further details on wavelet analyses and GAMs are provided below.

### 2.4.1 Wavelet coherence and phase

We applied the continuous Morlet wavelet transform $W_\sigma(t)$ (Addison, 2002) to each time series following approaches outlined
by Sheppard et al. (2016, 2017, 2019). The wavelet transform $W_\sigma(t)$ uses small localized wavelike functions to transform time
series signals into a more useful form (Addison, 2002). For this study, scaling was applied such that one wavelet oscillation





was equal to two months ($\sigma = 2$) because a two-month period is the highest-frequency fluctuation that can be identified in monthly time series. Wavelets were generated across a range of timescales, from two to 120 months (10 years). Wavelet transforms require scalloping to remove poorly estimated values at the tails of time series (Addison, 2002). For a 30-year time series of monthly observations, scalloping limits wavelet periods to a maximum of ~120 months, or approximately one-third of the length of the full time series (e.g., Fig. S1).

Coherence between two variables is a measure of the strength of association between those variables in a timescale-specific way that is not confounded by lagged or phase-shifted associations (Reuman et al., 2021). Accordingly, wavelet coherence quantifies the degree to which two time series have correlated magnitudes of oscillation and consistent phase differences through time, as a function of timescale (Walter et al. 2021), with magnitude ranging from 0 (no relationship) to 1 (perfect coherence). The 'wavelet mean field' normalization method (Sheppard et al. 2016) was used to measure wavelet coherence

because its coherence magnitude increases both with increasing synchrony between time series and when oscillations in time series at time $t$ and timescale $\sigma$ have similar phase (direction and time-lag). This method also permits testing for significant coherence across multiple timescales. Two timescale bands were tested for significance. Short timescales ($\leq 18$ months) were selected based on Buffalo Pound Lake intra-annual/seasonal dynamics and potential lagged relationships between DOC and environmental predictors. Long timescales ($> 18$ months—i.e., up to 120 months) were selected based on multi-year to decadal

patterns observed in DOC concentration.

Phase (direction and time-lag) was also investigated for environmental predictors that were significantly coherent with DOC concentration. Coherent variables may be in-phase (positively correlated) or anti-phase (negatively correlated), and are typically time-lagged (positively or negatively). To understand phase difference relationships between significantly coherent DOC–predictor pairs we computed the average phase $\phi$ across the corresponding timescale band (Walter et al., 2021). Because

phase $\phi$ is an angular measurement, cosine- or sine-transforming this value provides information about how close the relationship is to being in-phase [$\cos(\phi)$], and whether the time-lagged relationship between time series tends to be positive or negative [$\sin(\phi)$] (Walter et al. 2021). Cosine transformation assigns in-phase relationships ($\phi = 0$) to 1, anti-phase relationships ($\phi = \pm \pi$) to –1, and quarter-phase (i.e., time-lagged) relationships ($\phi = \pm \pi/2$) to 0. Sine transformation assigns both in-phase ($\phi = 0$) and anti-phase ($\phi = \pm \pi$) relationships to a value of 0 because they exhibit no time lag. When a change in DOC leads

ahead of a predictor variable ($\phi = -\pi/2$) the relationship is lagged negative, whereas when a change in DOC lags behind a change in a predictor ($\phi = \pi/2$) the relationship is lagged positive.

To test for significant coherence, all time series were transformed using standard optimal Box-Cox normalization prior to wavelet transformation (Sakia, 1992). Box-Cox transformation improves normality and ensures variability in individual time series is not dominated by extreme values (Sheppard et al., 2019). Significance testing for wavelet methods relies on Fourier

surrogate techniques where time series are normally distributed (Schreiber and Schmitz, 2000), so fair comparisons and statements of significance can only be made if underlying data are normally distributed (Sheppard et al., 2019). Box-Cox transformation removes the linear trend for each time series and re-scales the variance to 1 producing transformed times series





with mean of 0 and approximately normal distributions. Fourier transform–based methods for generating surrogate coherence data sets can be used to test for statistically significant coherence relationships between wavelet-transformed variables

(Sheppard et al., 2017). These Fourier transformed data retain the original characteristics (e.g., temporal autocorrelation) of the time series and test whether coherence values are likely to occur under the null hypothesis that no actual coherence and phase relationships exist between variables. For each DOC–predictor time series pair 10,000 surrogate randomizations were run to facilitate more accurate significance testing results and reduce variability on repeat runs (Reuman et al., 2021; Sheppard et al., 2017).

## 2.4.2 Generalized additive models

Predictors identified as significant by coherence analysis were used to model the DOC time series using a GAM. This approach was chosen because it can account for nonlinearity in trends whereas other methods are limited to identifying increasing or decreasing linear (monotonic) trends (e.g., (seasonal) Mann-Kendall test) or require *a priori* selection of the functional form of trends in time series or selection from less flexible polynomial models (e.g., parametric linear or generalized linear models)

(Finlay et al., 2019; Simpson, 2018). GAM's use of splines is also advantageous in that it can reduce bias issues and over-fitting at the tails of data sets, which is a common problem with polynomial models (Finlay et al., 2019). In practice a GAM takes the general form:

$$y_i = \beta_0 + f_1(x_{i1}) + f_2(x_{i2}) + \cdots + f_n(x_{in}) + \epsilon_i \,, \tag{5}$$

where $y_i$ is the response variable; $\beta_0$ is the model intercept; $x_{i1}$, $x_{i2}$, and $x_{in}$ are covariates; $f_1$, $f_2$, and $f_n$ are non-parametric

smoothing functions; and $\epsilon_i$ are independent $N(0, \sigma^2)$ random errors (Wood, 2017). All predictors (i.e., $f(x_i)$) were estimated using thin plate regression splines and penalized using restricted maximum likelihood–based smoothness selection procedures (Wood, 2011). Initial basis dimension ($k$) of each smooth function was checked following the procedure described in Pya and Wood (2016)—if initial $k$ was deemed too low (i.e., if $k$-index < 1 and estimated degrees of freedom was close to $k'$), a larger basis size was used and the model refitted. The Tweedie distribution (Tweedie, 1984) was used as the model conditional

distribution because histograms of most of the environmental predictors showed deviation from normality and resembled some form of the gamma distribution, which is also included in the Tweedie family of distributions and identified when the Tweedie power parameter, $p$, is 2. In our model $p$ was 1.99, indicating the gamma distribution was a good fit; however, selecting the Tweedie distribution *a priori* avoided the potential of erroneously choosing gamma as the conditional distribution. Model fit was assessed through qualitative inspection of quantile–quantile residuals, residuals vs. linear predictor, histogram of residuals,

and observed vs. fitted values plots. Uncertainty in the estimated DOC trend was simulated using 10,000 iterations from the posterior distribution of the fitted values. The simulated trends are consistent with the estimated trend but include the uncertainty in the estimates of the spline coefficients (Finlay et al., 2019). The posterior simulation involves drawing 10,000 samples from the multivariate normal distribution then deriving the difference between peak and minimum DOC for each



sample (trend). The upper and lower 2.5% probability quantiles of the distribution of 10,000 differences in trend for each year
form a 95% confidence interval on the difference estimated from the fitted trend (Finlay et al., 2019).

## 3 Results

### 3.1 Temporal flow and water chemistry parameters

Flow sources varied over the 30-year period with no distinct long-term or multi-year patterns (Fig. 2a–c). Buffalo Pound Lake
inflows ($Q_{BP}$) and $Q_{LD}$ each showed seasonal patterns, peaking around late spring to early summer and reduced during winter
months when water demands are lower and the lake is covered with ice. Local catchment flows ($Q_{LC}$) typically peaked in late
spring to early summer and did not contribute water to Buffalo Pound Lake in winter months or dry years. Streamflow averaged
$3.2 \pm 2.6$ (range: 0–12.4) m$^3$ s$^{-1}$ at $Q_{LD}$, and $3.6 \pm 2.8$ (range: 0.1–18.9) m$^3$ s$^{-1}$ at $Q_{BP}$ (Table S3). Catchment flows were much
lower, averaging $0.3 \pm 0.8$ (range: 0–8.3) m$^3$ s$^{-1}$. In years where $Q_{LC}$ was low or absent, particularly in the late 1990s and early
2000s, $Q_{BP}$ generally followed patterns of water release from Lake Diefenbaker and fluctuations in $Q_{BP}$ were comparable to
those of $Q_{LD}$. Several years showed evidence of wetter-than-normal conditions (e.g., 1997, 2014) with short-term episodes of
considerable discharge from the local catchment. These episodes correspond to peaks in $Q_{BP}$ and were generally associated
with low $Q_{LD}$ flows (Fig. 2).

Dissolved organic carbon concentrations fluctuated considerably over the 30-year observation period, ranging 3.3–12.4 mg L$^{-1}$ (Fig. 3, Table S3). Mean ($\pm$ standard deviation) DOC concentration was $6.8 \pm 1.8$ mg L$^{-1}$. Several notable fluctuations
occurred over multi-year periods. For example, after sustained high concentrations from 1997–2000 (mean 9.5 mg L$^{-1}$), DOC
fell to 4.4 mg L$^{-1}$ by 2004 and remained near- or below-average until the mid-2010s. Between 1990 and 2000, there was a
gradual increase in DOC concentration where, alongside intraannual increases and decreases, DOC concentrations increased
from 3.9 mg L$^{-1}$ in 1990 to 9.3 mg L$^{-1}$ in 2000. Other notable changes occurred at shorter timescales: in 1991 DOC
concentrations were $\leq 5.0$ mg L$^{-1}$ from January to July, spiked to 10.5 mg L$^{-1}$ in August, and fell to 5.5 mg L$^{-1}$ by December.
The mid-2010s also saw a sharp increase in DOC concentration followed by below-average levels within a 4-year span.

Water chemistry was also highly variable over the 30-year period (Fig. 2d–i, Table S3), with evidence of dramatic changes in
the chemistry of Buffalo Pound Lake over time. For example, $SO_4^{2-}$ concentrations ranged 56.8–340 mg L$^{-1}$ and were relatively
stable from 1990–2009 before rapidly rising up to 250 mg L$^{-1}$ between 2011 and 2017 (Fig. 2d). By the end of 2019, $SO_4^{2-}$
returned to levels observed during 1990–2010. Nitrate also showed two distinct patterns. After averaging $0.20 \pm 0.28$ mg L$^{-1}$
from 1990–1999 and peaking at 1.5 mg L$^{-1}$ in 1997, $NO_3^-$ concentrations were nearly an order of magnitude lower from 2000–
2019, averaging $0.05 \pm 0.08$ mg L$^{-1}$ (Fig. 2h). Total phosphorus, SRP, Chl $a$, and $NH_4^+$ concentrations rose rapidly in 1991
(Fig. 2e–g, i), concomitant with the rise in DOC in that year, but this was transient. In the few years after 1991, chemical
concentrations were average or below average, followed by increases in concentrations into the early 2000s. The mid 2010s
also showed elevated levels of $SO_4^{2-}$, TP, Chl $a$, and $NH_4^+$ concurrent with elevated DOC concentrations and $Q_{LC}$ during this
time.




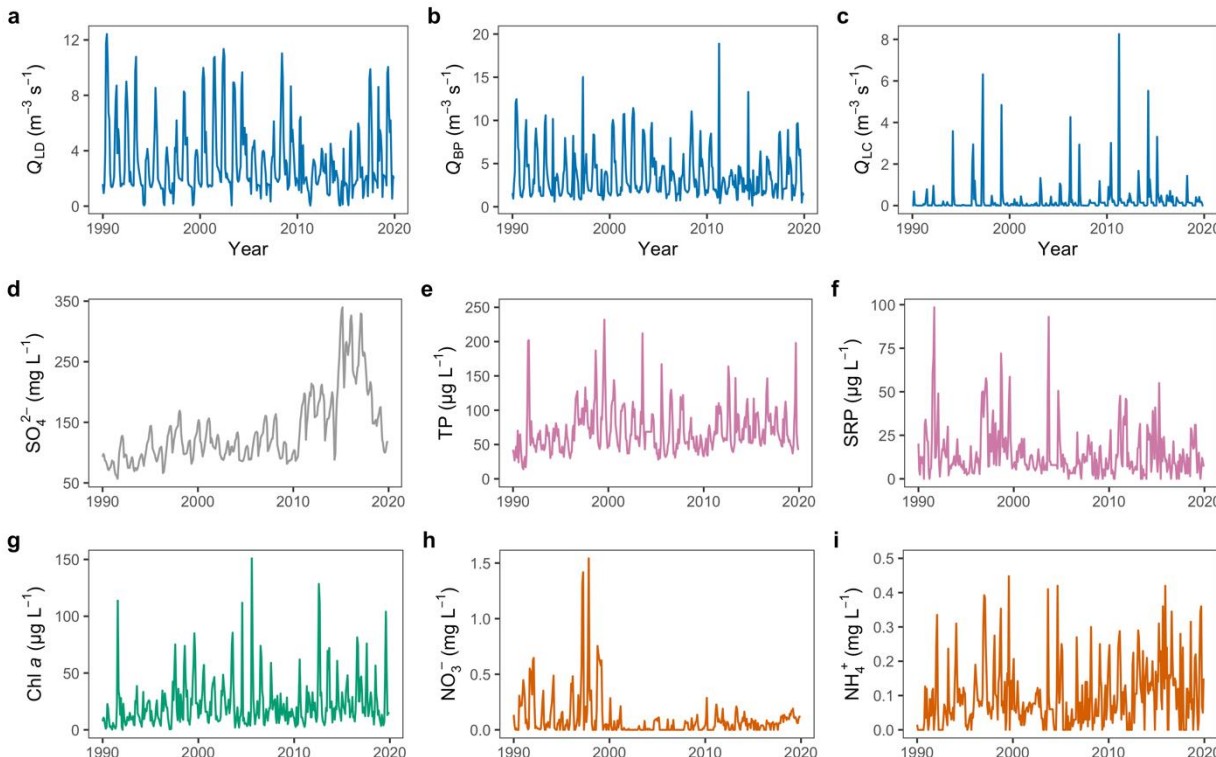

**Figure 2. (a–i) Monthly time series of flows from $Q_{LD}$, $Q_{BP}$, $Q_{LC}$, and SO$_4^{2-}$, TP, SRP, Chl *a*, NO$_3^-$, and NH$_4^+$, concentrations from 1990–2019. Note that y-axis scales for $Q_{LD}$, $Q_{BP}$, and $Q_{LC}$ plots are different. (All N concentrations are as mg N L$^{-1}$).**



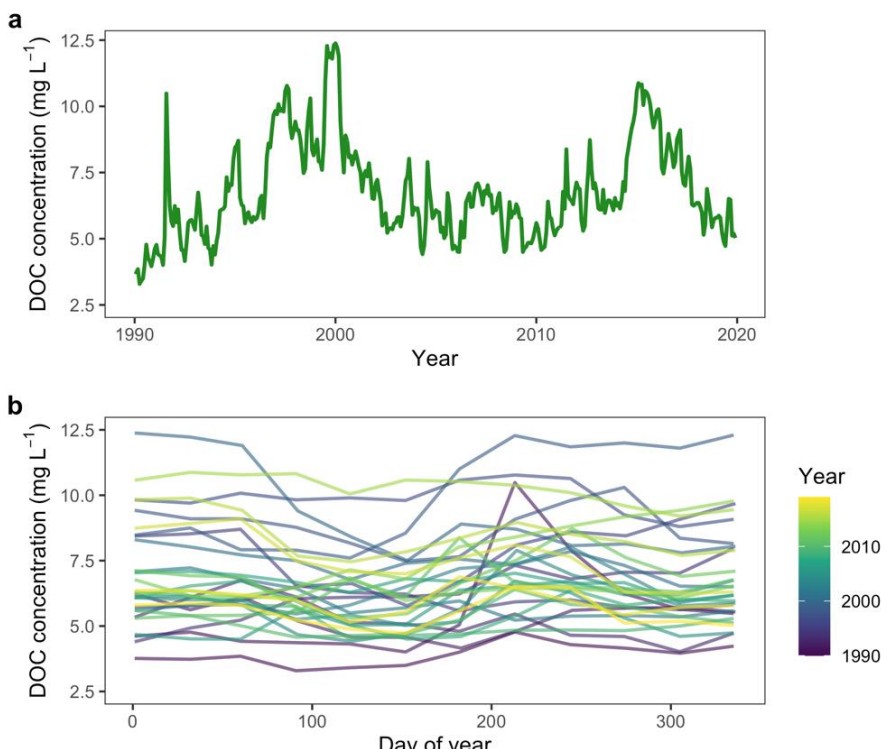

**Figure 3. Monthly Buffalo Pound Lake DOC concentrations from 1990–2019 by (a) year, showing the pattern in DOC concentration over the full time series, and (b) day of year, showing the intraannual variation in DOC concentration.**





## 3.2 Wavelet coherence and phase

Coherences between DOC and nine environmental predictors at short ($\leq$ 18-month) and long (> 18-month) timescales were
highly variable, ranging from 0.05–0.74 (median of 0.31) (Table 1, Fig. S2). At short timescales coherence magnitudes ranged
0.08–0.50 (median of 0.15) and at long timescales 0.05–0.74 (median of 0.54). Four DOC–predictor pairs were significant at
short timescales ($\alpha$ = 0.05 significance level): $SO_4^{2-}$, TP, Chl *a*, and $NH_4^+$. At long timescales $SO_4^{2-}$, TP and $Q_{LD}$ were
significantly coherent with DOC ($\alpha$ = 0.05 significance level). We observed generally higher coherence magnitudes at long
timescales compared to short timescales (Table 1); this is consistent with potential bias in wavelet coherence where greater
coherence magnitudes tend to be returned at long timescales (Walter et al., 2021).

Phase relationships spanned the range of possible values (0 to ± $\pi$) (Fig. S3). For DOC–predictor pairs with significant
coherence, most relationships were approximately in-phase ($-\pi/4 < \phi < \pi/4$) at both short and long timescales (Table 1). At
short timescales DOC was strongly in-phase with TP, Chl *a*, and $NH_4^+$ ($0 < \phi < \pi/4$), and tended to lag behind fluctuations in
$SO_4^{2-}$ ($-3\pi/4 < \phi < -\pi/2$). Interestingly, the phase relationship between DOC and $SO_4^{2-}$ shifted to strongly in-phase ($-\pi/4 < \phi$
< 0) at long timescales, and TP remained strongly in-phase with DOC ($0 < \phi < \pi/4$). Unlike $SO_4^{2-}$ and TP, fluctuations in DOC
tended to lag behind fluctuations in $Q_{LD}$ ($-3\pi/4 < \phi < -\pi/2$).



**Table 1. Coherence and phase relationships between DOC and environmental predictors at short (≤ 18-month) and long (> 18-month) timescales. Mean coherence $p$-values are calculated over the reported timescale band for each significant relationship. Significant relationships are bolded and denoted by \*\*\* ($p < 0.001$), \*\* ($p < 0.01$), and \* ($p < 0.05$). Relationships marginally non-significant ($p < 0.1$) are italicized and denoted by †. Cos(ϕ) and sin(ϕ) are transformations of the average phase ϕ over the reported timescale band where cos(ϕ) describes how close the relationship between DOC and each parameter is to being in-phase and sin(ϕ) focuses on whether the time-lagged relationship between DOC and each predictor tends to be positive or negative. Because phases are angular measurements, ϕ ranges between −π and π, and cos(ϕ) and sin(ϕ) range between −1 and 1. Relationships are identified as in-phase when cos(ϕ) ≈ 1, anti-phase when cos(ϕ) ≈ −1, and quarter-phase when cos(ϕ) ≈ 0. When sin(ϕ) ≈ −1 the relationship is time-lagged positive (i.e., a change in DOC precedes a change in the predictor), whereas when sin(ϕ) ≈ 1 the relationship is time-lagged negative (DOC lags behind the predictor). When relationships are perfectly in-phase or anti-phase (i.e., not time-lagged), sin(ϕ) = 0.**

| Environmental predictor | Timescale | Mean coherence | ϕ | cos(ϕ) | sin(ϕ) | Phase relationship |
|---|---|---|---|---|---|---|
| $SO_4^{2-}$ | Short | **0.21\*** | −2.01 | −0.43 | −0.90 | DOC lags behind $SO_4^{2-}$ |
| TP | Short | **0.36\*\*** | 0.33 | 0.95 | 0.33 | TP very strongly in-phase with DOC |
| SRP | Short | 0.08 | −0.93 | 0.60 | −0.80 | — |
| Chl $a$ | Short | **0.50\*\*\*** | 0.55 | 0.85 | 0.52 | Chl $a$ strongly in-phase with DOC |
| $NH_4^+$ | Short | **0.27\*\*** | 0.16 | 0.99 | 0.16 | $NH_4^+$ very strongly in-phase with DOC |
| $NO_3^-$ | Short | 0.15 | 0.38 | 0.93 | 0.37 | — |
| $Q_{LD}$ | Short | 0.10 | −3.02 | −0.99 | −0.12 | — |
| $Q_{BP}$ | Short | 0.12 | 2.98 | −0.99 | 0.16 | — |
| $Q_{LC}$ | Short | 0.10 | 2.62 | −0.87 | 0.50 | — |
| $SO_4^{2-}$ | Long | **0.67\*\*** | −0.22 | 0.98 | −0.22 | $SO_4^{2-}$ very strongly in-phase with DOC |
| TP | Long | **0.57\*** | 0.45 | 0.90 | 0.43 | TP very strongly in-phase with DOC |
| SRP | Long | 0.54 | 0.89 | 0.63 | 0.77 | — |
| Chl $a$ | Long | 0.34 | 0.39 | 0.93 | 0.38 | — |
| $NH_4^+$ | Long | 0.05 | 1.36 | 0.21 | 0.98 | — |
| $NO_3^-$ | Long | 0.15 | 1.89 | −0.31 | 0.95 | — |
| $Q_{LD}$ | Long | **0.74\*\*** | −1.84 | −0.26 | −0.96 | DOC lags behind $Q_{LD}$ |
| $Q_{BP}$ | Long | *0.66†* | −2.13 | −0.53 | −0.85 | DOC lags behind $Q_{BP}$ |
| $Q_{LC}$ | Long | 0.52 | 1.53 | 0.04 | 1.00 | NA |



### 3.3 Generalized additive modelling

Five environmental predictors ($SO_4^{2-}$, TP, Chl $a$, $NH_4^+$, and $Q_{LD}$) found to be significantly coherent with DOC concentration as identified by wavelet coherence analyses at either short ($\leq$ 18-month) or long ($>$ 18-month) timescales were included in a simple GAM. A thin plate regression spline for each covariate explained 56% of the deviance in DOC concentration over the 30-year period, with an adjusted $R^2$ of 0.50 (effective degrees of freedom = 16.9). Sulfate, TP, $NH_4^+$, and $Q_{LD}$ all explained significant variation in DOC concentration ($p < 0.001$ for $NH_4^+$ and $Q_{LD}$, and $p < 0.0001$ for $SO_4^{2-}$ and TP). The Chl $a$ smooth

term was the only non-significant predictor ($p = 0.15$; Fig. 4a). Total phosphorus and $Q_{LD}$ showed linear positive and negative relationships with DOC concentration respectively (Fig. 4b,e). At $NH_4^+$ concentrations $<$ ~0.1 mg L$^{-1}$ (61% of the observed $NH_4^+$ data), $NH_4^+$ had an approximately linear positive effect on DOC concentration but did not have much predictive power at concentrations $>$ 0.1 mg L$^{-1}$ as shown by the increasingly large confidence interval (CI; shaded region) at higher $NH_4^+$ levels (Fig. 4c). The relationship between DOC and $SO_4^{2-}$ (Fig. 4d) was generally positive but was more variable than the other

environmental drivers. For example, at $SO_4^{2-}$ concentration from ~150–200 mg L$^{-1}$ increases in $SO_4^{2-}$ were concurrent with decreases in DOC; however, at $SO_4^{2-}$ concentrations $<$ 150 mg L$^{-1}$ (71% of the observed $SO_4^{2-}$ data) DOC tended to increase with increases in $SO_4^{2-}$. At higher concentrations of $SO_4^{2-}$, TP, and $NH_4^+$ and high $Q_{LD}$ the GAM lost predictive power (large CI for right tails of each smooth) but performed well in lower to moderate ranges (Fig. 4). The GAM provided a good fit between observed and fitted DOC at concentrations $<$ ~7 mg L$^{-1}$ but considerable variation was apparent in the model fit at

concentrations $>$ 8 mg L$^{-1}$ (Fig. S4). Fitting the GAM to the DOC time series (Fig. 5) revealed that using these predictors to estimate DOC concentration was useful at identifying multi-year trends (e.g., in the 1990s and mid-2010s) but suffered from over-fitting intra-annual fluctuations.



**Figure 4. Partial effects of (a) Chl *a*, (b) $Q_{LD}$, (c) $NH_4^+$, (d) $SO_4^{2-}$, and (e) TP for the GAM fitted to the DOC time series with**
**significantly coherent environmental predictor covariates. The *x*-axes show observed values for each environmental predictor. The *y*-axes show the partial effects of thin plate regression spline smooths (black lines) for each environmental predictor. Grey shaded regions are the 95% confidence interval. The rug (inset on *x*-axis) displays the distribution of observed values. Chlorophyll *a* was the only non-significant covariate (*p* = 0.15); all other environmental predictors were significant at *p* < 0.001.**





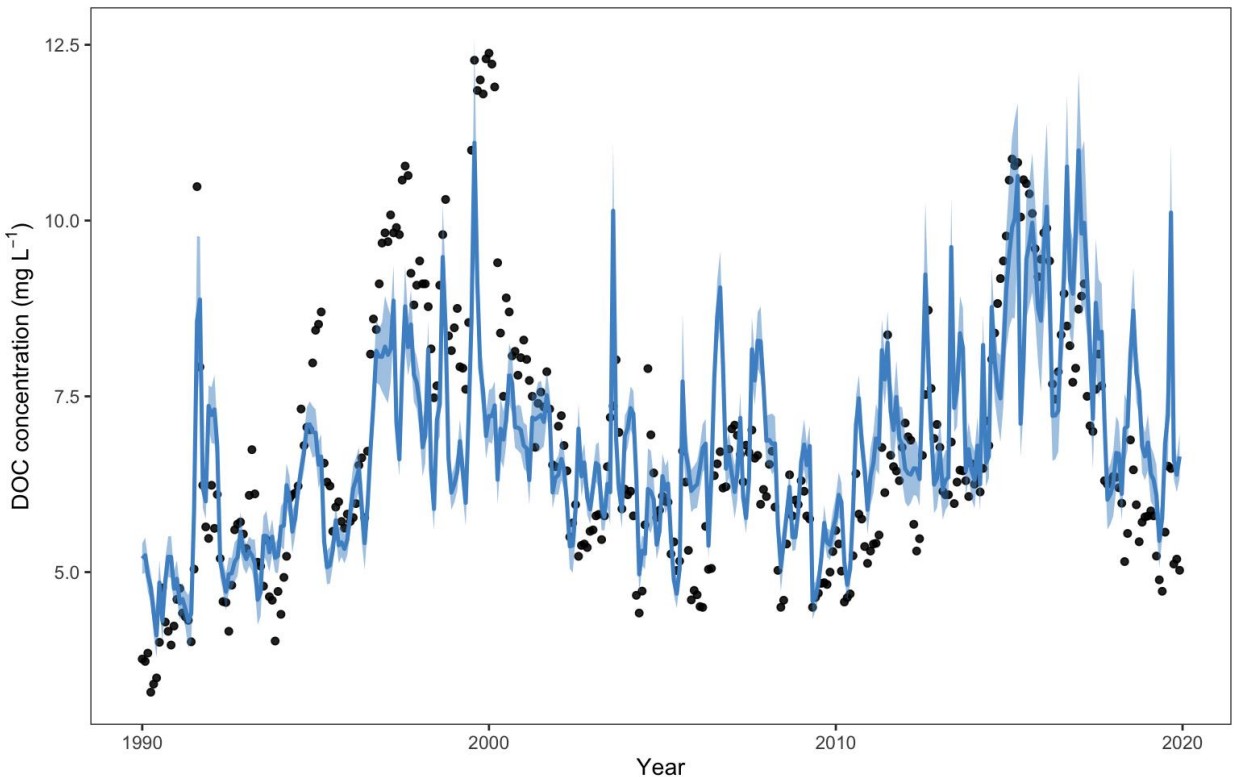

**Figure 5. DOC concentration time series (black points) overlain by the GAM fit (GAM model using $SO_4^{2-}$, TP, Chl $a$, $NH_4^+$, and $Q_{LD}$; blue line). The shaded blue area is the 95% confidence interval of the GAM model fit using 10,000 samples drawn from the posterior multivariate normal distribution. This model explained 56% of the deviance in DOC concentration with an $R^2_{adj}$ of 0.50.**



## 4 Discussion

### 4.1 Variability in DOC concentration and flow conditions

Globally, numerous drivers of changing DOC concentrations have been identified, from declining atmospheric $SO_4^{2-}$ (Monteith et al., 2007) and N deposition (Evans et al., 2008), to land management and land-use (Yallop and Clutterbuck, 2009), and to patterns in temperature, precipitation, or runoff (Hongve et al., 2004; Weyhenmeyer and Karlsson, 2009). Within Buffalo Pound Lake, a shallow reservoir in a DOC-rich landscape (Labaugh et al., 1987; Waiser, 2006) with high climatic variability (Pham et al., 2009; Pomeroy et al., 2007; Vogt et al., 2018), climate and flow source appear to have an overriding

influence on DOC concentrations. In this highly flow-managed lake, management decisions are related to weather and climatic patterns. In dry years more water is released from upstream Lake Diefenbaker to maintain water levels at Buffalo Pound Lake and other lakes downstream. In wet years runoff from the local catchment increases, delivering nutrients, solutes, and DOC from prairie croplands and wetlands to Buffalo Pound Lake through tributaries like Ridge Creek and Iskwao Creek. Dry years constitute a more "managed" flow regime (with high $Q_{LD}$) whereas wet years bring the system closer to "natural" flow

conditions. Under the "natural" regime allochthonous DOC influxes may be large, as prairie wetlands can be important flow sources during wet periods when hydrologic connectivity is high (Ali and English, 2019; Nachshon et al., 2014) and these prairie wetlands can have extremely high DOC concentrations due to evapoconcentration (in excess of 100 mg L$^{-1}$ in some ponds) (Arts et al., 2000; Waiser, 2006). Wetlands are not the only substantive DOC source in the landscape. High concentrations of DOC have also been observed in snowmelt runoff from agricultural fields in the prairies. For example, flow-

weighted mean DOC concentrations in snowmelt runoff under conventional and rotational tillage practices in the prairies are also relatively high: 15.8 mg L$^{-1}$ under conventional tillage and 20.0 mg L$^{-1}$ with rotational tillage (Liu et al. 2014). In addition to soils, crop residues can be an important DOC source in the prairies, but DOC release from residues may be highly dependent on crop type (Elliott et al., 2008; Elliott, 2013).

Lake DOC concentrations in cold regions tend to vary both seasonally and interannually. Seasonal cycles alter the rates of

biological DOC production and hydrological export (Clark et al., 2010), and DOC influxes often peak during spring snowmelt while biological DOC production peaks in summer (Buffam et al., 2007; Clark et al., 2010; Laudon et al., 2004). In cold regions where lakes are ice covered for a portion of the year, DOC exclusion from ice can be important for driving winter metabolism and DOC concentrations (Guo et al., 2012; Kurek et al., 2022), whereas photo- and biodegradation processes (Hansen et al., 2016) and autochthonous DOC production (Baron 2023) may dominate during the open-water season. In many

lakes seasonal variation in DOC concentration can be one to two orders of magnitude greater than the long-term rate of change over decadal time scales (Clark et al., 2010). Within Buffalo Pound Lake we observed both high inter- and intra-annual variability in DOC concentration, with large changes occurring over multi-year periods (e.g., up to a 10 mg L$^{-1}$ increase in monthly concentrations over a decade) and fluctuation as high as 7 mg L$^{-1}$ within a single year. This lake has a short residence time (8 months to 2.5 years; BPWTP, 2016; Vogt et al., 2018), which may make it more responsive to rapid change as a result



of changes in inflows. As well, both allochthonous and autochthonous sources are important to DOC in Buffalo Pound Lake (Baron 2023). Variable residence time and the complex suite of conditions affecting DOC can create challenges in identifying factors driving DOC concentration. Here, we use wavelet analysis and GAMs to understand these observed large within- and among-year fluctuations in DOC.

## 4.2 DOC–predictor relationships

Wavelet analyses demonstrated that $Q_{LD}$ was coherent with DOC concentrations in Buffalo Pound Lake at long timescales, with these flows tending to precede fluctuations in DOC concentration. Specifically, $Q_{LD}$ appears to have a flushing, or diluting impact. Lake Diefenbaker, with low DOC concentrations, is fed by the South Saskatchewan River, a sand-bottom river whose origins are in the Rocky Mountains. We expected to see a contrasting role of the local catchment. Specifically, given that allochthonous DOC can increase rapidly in lakes during periods of elevated runoff (e.g., during snowmelt or large precipitation

events) and prairie wetlands generally have DOC concentrations much higher than Buffalo Pound Lake, we expected $Q_{LC}$ to be coherent with DOC at short timescales. Instead, $Q_{LC}$ coherence with DOC was very low (0.10) and non-significant at short timescales (Table 1). This result likely reflects the generally smaller contribution of the local catchment to flow and intermittency of these sources making signals challenging to ascertain. Local catchment flows, though sometimes large, are infrequent (Fig. 2) due to the ephemeral nature of these sources and their tendency to have no or minimal flow in drier years.

Thus, characterizing coherence for this intermittent flow is more challenging. Large pulses of inflows from the local catchment do still appear important to the lake and the local catchment is likely an important source of allochthonous DOC when $Q_{LC}$ occurs (see below). Flows into Buffalo Pound Lake were likewise not coherent with DOC at both short and long timescales (Table 1). Because $Q_{BP}$ combines flows from Lake Diefenbaker and parts of the local catchment, this metric may obscure relationships by integrating different mechanisms by which the two flow sources impact chemistry (e.g., greater allochthonous

inputs from the local catchment, and more dilute waters from Lake Diefenbaker).

In addition to $Q_{LD}$, $SO_4^{2-}$, TP, Chl $a$, and $NH_4^+$ were also significantly coherent with DOC at short and/or long timescales. At short timescales ($\leq$ 18 months) DOC lagged behind $SO_4^{2-}$ on average but at long timescales (> 18 months) this relationship was strongly in-phase, suggesting within-year or seasonal DOC and $SO_4^{2-}$ dynamics may differ, but their patterns become more synchronous at longer timescales. Total phosphorus, Chl $a$, and $NH_4^+$ concentrations were strongly in-phase with DOC

at short timescales, and TP remained strongly in-phase at long timescales. Possible mechanisms for these relationships include internal production and transformations, and synchronous inputs of nutrients and allochthonous DOC from the local catchment (see below).

Using $SO_4^{2-}$, TP, $NH_4^+$, Chl $a$, and $Q_{LD}$ as predictors in a GAM describing DOC explained 56% of the deviance in DOC concentrations observed over this 30-year data set, where all predictors were significant except for Chl $a$. This result presents

evidence that DOC concentration in Buffalo Pound Lake is strongly linked to its major flow source, Lake Diefenbaker, and in-lake nutrient and solute chemistry. We attempted to incorporate time-lagged relationships between DOC and $SO_4^{2-}$ and $Q_{LD}$ into the GAM approach; however, a feature of wavelet coherence analysis is that phase relationships can only be determined





*on-average* over the timescale of interest (J. Walter pers. comm.), and thus specific time-lags (e.g., 6 months, 24 months, etc.) cannot be accurately measured with this approach. The true time-lag relationship between $Q_{LD}$ and DOC also likely varies

depending on flow magnitude. To assess concerns over potential collinearity between DOC and TP we ran a GAM with SRP in place of TP but found no discernable differences in model results.

Dissolved organic carbon was not the only chemical variable that underwent dramatic change within Buffalo Pound over the study period. Indeed, 6-fold variation in concentration of $SO_4^{2-}$ was observed. The rapid rise and fall in DOC and $SO_4^{2-}$ concentrations in the 2010s may be linked to both internal and external processes. The majority of $SO_4^{2-}$ fluxes to lakes come

from soils and wetlands in the catchment, often in mineral forms or bound in allochthonous organic matter containing oxygen-bound S (e.g., as ester-$SO_4$ or other organo-$SO_4$ species) (Couture et al., 2016), helping explain coherence between $SO_4^{2-}$ and DOC. Much like prairie catchments have high DOC, they can also have high $SO_4^{2-}$, again, due to soil and wetland pools. Pothole ponds have a wide range of $SO_4^{2-}$ concentrations, with a median of 163 (mean 519) mg L$^{-1}$, and some ponds reaching concentrations of 5500 mg L$^{-1}$ (L. Dyck unpublished data for ~150 prairie pothole region ponds in 2019) and prairie soils can

be similarly high in $SO_4^{2-}$ (Fennell and Bentley, 1998). Annually-measured $SO_4^{2-}$ samples at Lake Diefenbaker from 1990 to 2019 showed median concentrations of 58 mg L$^{-1}$ and maxima < 90 mg L$^{-1}$ (J.-M. Davies unpublished data), values near the low range of $SO_4^{2-}$ seen in Buffalo Pound Lake. The much higher median Ridge Creek $SO_4^{2-}$ (estimated at ~1094 mg L$^{-1}$; unpublished data), combined with below-average $Q_{LD}$ (and higher $Q_{LC}$) in the early 2010s mean that catchment $SO_4^{2-}$ sources must have contributed to the higher Buffalo Pound Lake $SO_4^{2-}$ concentrations during the latter part of the observation record.

Extreme flooding across the prairies in 2011 could have created an exceptional 'fill and spill' scenario (see Nachshon et al., 2014; Shook and Pomeroy, 2012) where water containing elevated levels of $SO_4^{2-}$ and DOC was transported via runoff from spilling pothole ponds to Buffalo Pound Lake. Similar mechanisms were likely important in 2014 and 2015, two years with periods of high precipitation in the region (Water Security Agency, 2018) and above-average catchment inflows, when $SO_4^{2-}$ and DOC concentrations further increased. There is also the potential that groundwater may be infusing minerals such as $SO_4^{2-}$

to the lake (BPWTP, 2022) but we lack data to support this $SO_4^{2-}$ source. Prairie soils and lake sediments rich in gypsum ($CaSO_4$) and pyrite ($FeS_2$) deposited during glaciation 10,000 years ago may also play a role. Reoxidation of lake sediments rich in $H_2S$, $CaSO_4$, $FeS_2$, and other sulfides can also increase surface water $SO_4^{2-}$ concentrations (Holmer and Storkholm, 2001), and in freshwater sediments, high rates of oxidation of reduced S compounds can shift the sediments from net sinks to sources of $SO_4^{2-}$ (Holmer and Storkholm, 2001), particularly if groundwater $SO_4^{2-}$sources to the lake are not high.

Strong in-phase coherence between DOC and TP at both short and long timescales, as well as the strong linear effect of TP on DOC (Fig. 4e), suggest that there is synchrony in the processes influencing DOC and TP in Buffalo Pound Lake. As with $SO_4^{2-}$, prairie ponds can also be important P sources to downstream lakes during periods of elevated catchment flow, particularly because ponds high in $SO_4^{2-}$ tend to have greater sediment P release rates (Jensen et al., 2009). Spring TP concentrations measured at ~150 ponds across the prairie region in 2019 revealed mean TP concentrations in excess of 500 µg P L$^{-1}$

(McFarlan, 2021) suggesting potential for notable nutrient inputs during periods of elevated $Q_{LC}$. It has also been suggested





that larger pulses of internal P released from sediments during transient stratification events could be linked to mid-summer algal blooms (D'Silva, 2017; Painter et al. 2022b), events which could trigger pulses of greater autochthonous production. Chlorophyll *a* concentration, a measure of algal abundance, was not significant in the GAM but showed in-phase coherence with DOC concentrations at short timescales, suggesting a seasonal effect associated with cyanobacterial/algal biomass in the

lake. Inefficient conversion of light energy to organic molecules during photosynthesis means that a portion of the organic materials produced by algal cells are released as DOM/DOC to ambient water over the lifetime of the algae (Fogg, 1966; Myklestad, 1995). Extracellular DOC release, via the overflow model (Fogg, 1966, 1983; Williams, 1990) or the passive diffusion model (Bjørnsen, 1988; Fogg, 1966), has been shown to be an important source of autochthonous DOC, where up to 5–35% of fixed organic carbon may be release immediately as DOM, often with high proportions of DOC (Carlson and

Hansell, 2015). Other mechanisms for release of autochthonous carbon, such as release by grazers, and following sedimentation, are also likely important (Keller and Hood, 2011 ). Additionally, DOC concentrations can increase by 1–2 mg L$^{-1}$ between the Buffalo Pound Lake inflow and outflow in years dominated by flow from Lake Diefenbaker (Baron 2023), consistent with autochthonous production, and the potential for extracellular release.

Observed in-phase coherence between DOC and NH$_4^+$ at short timescales and the significance of NH$_4^+$ as a predictor of DOC

in the GAM may be related to several mechanisms. The lake is sometimes N-limited (Swarbrick et al., 2019), suggesting increased NH$_4^+$ could support increased productivity, and autochthonous production. This temperate lake is also highly seasonal, with major summer blooms (Painter, Venkiteswaran, and Baulch, 2022; Painter, Venkiteswaran, Simon, et al., 2022), followed by long, cold winter periods when the lake is ice-covered and NH$_4^+$ dynamics undergo rapid change (Cavaliere and Baulch, 2020). The early winter phase is marked by increasing NH$_4^+$ concentrations associated with decreased autotrophy,

oxygen depletion, and organic matter mineralization. We observed several years where DOC concentrations decreased during the early winter period. The late winter phase shows a decline in NH$_4^+$ likely associated with biotic uptake. Seasonal variation in DOC:DON ratios are also apparent (Fig. S5). Higher C:N occurs under ice cover then decreases beginning around spring ice-off (~100[th] day of year), reaching its minima in late summer concurrent with seasonal maximum Chl *a* concentrations.

### 4.3 Flow management and effective water treatment

Cyclic wet–dry phases in the prairie region are associated with elevated flood risk and severe drought. Examples include the major drought from 1999–2004 and extensive flooding in 2011 and 2014. Flood risk and drought underpin water management concerns in the region, which include balancing the provision of water from Buffalo Pound Lake to meet the needs of more than 280,000 people. "Natural" and "managed" flow scenarios associated with wet–dry cycles each present challenges for Buffalo Pound Lake water quality and for the Buffalo Pound Water Treatment Plant (BPWTP). Years with high local

catchment flows (natural flow scenario) lead to high DOC and poor water quality that can take years to dissipate. For example, during a prolonged drought from 1999–2005 specific ultraviolet absorbance at 254 nm normalized to DOC concentration (SUVA$_{254}$) values were very low (~0.9–1.6 L mg-C$^{-1}$ m$^{-1}$ over this period) but after extensive flooding in 2011, SUVA$_{254}$ exceeded 2.5 L mg-C$^{-1}$ m$^{-1}$ and remained above 1.5 mg L mg-C$^{-1}$ m$^{-1}$ into 2019 (Fig. S6). At the same time, managed flow



conditions—that arise when the system is dry, and flows from Lake Diefenbaker are important to maintaining water levels in
Buffalo Pound Lake—may contribute to worse algal blooms in Buffalo Pound Lake in terms of magnitude (Painter,
Venkiteswaran, and Baulch, 2022), and coincide with autochthonous DOC production in the lake (e.g., Baron 2023). Periods
of elevated DOC under either flow scenario can create added challenges and costs for the BPWTP to maintain safe drinking
water standards. At high DOC levels, water treatment is impaired by costly increases in coagulant loads required for pre-
treatment DOM removal (Cooke and Kennedy, 2001) and coliform regrowth in distribution systems (LeChevallier et al., 1996).
Untreated DOC can also contribute to poor taste and odour problems (Matilainen et al., 2011). When chlorine reacts with DOC
during water treatment, the compounds that comprise DOC can act as precursors for a suite of harmful disinfection byproducts
(DBPs) with potentially carcinogenic and mutagenic properties (Chow et al., 2003) requiring strict regulation (Health Canada,
2006, 2008). Disinfection byproduct formation is a major concern for the BPWTP (Williams et al., 2019), particularly when
DOC is high. Under these circumstances, the BPWTP must use a pre-chlorination step to reduce algal growth and prevent
rising floc (organic materials that coalesce and resist coagulation) that can impact treatment capacity by accumulating on
filtration media (Painter, Venkiteswaran, and Baulch, 2022). Pre-chlorination is typically required in the summer months when
algae are abundant in Buffalo Pound Lake (Painter, Venkiteswaran, and Baulch, 2022), and thus makes it difficult for the
BPWTP to meet regulatory limits for trihalomethanes in years where DOC is also elevated (BPWTP, 2016).

Environmental flows represent a framework that could be used to better manage competing needs for drinking water, as well
as the hydrological functions of the reservoir, and its critical fish and wildlife habitat.  The variable magnitude, timing, and
water quality of flows from two water sources and impacts on DOC underscores some of the complexity that would accompany
development of environmental flow rules for Buffalo Pound Lake. Maintaining stable water levels within the lake to limit
valley slumping and erosion is important (Saskatchewan Environment and Resources Management, 2001), and competing
regional priorities for freshwater, including hydropower production, flood control, and irrigation are also key factors in water
management (Water Security Agency, 2013; Wheater and Gober, 2013). Water quality concerns are broad, and include
increasing nutrient inputs to agricultural lands, agricultural drainage of wetlands which can have major impacts on catchment
flow (Spence et al., 2022), effluent disposal practices on agricultural land, leaking septics, and pollution from tourism and
recreational activities. Ultimately, concurrent efforts to consider environmental flow rules and advance sourcewater protection
could help ensure maximum benefit is attained from current investments, including a $325M investment in upgraded water
treatment (B. Kardash pers. comm.).

## 5 Conclusions

High variability in DOC concentration occurs at both long and short timescales in a shallow eutrophic reservoir. Flow source,
strongly influenced by climatic and hydrologic variability, is an important control on DOC concentration. Pulses of
allochthonous DOC from the local catchment during wet periods, and autochthonous production are linked to higher DOC in
the reservoir. Total phosphorus and $NH_4^+$ concentrations are synchronous with DOC and in-phase, suggesting that autochthony



as well as seasonal nutrient and organic matter dynamics (ice-covered, open-water seasons) are important in years where allochthonous inputs are lower. Sulfate, which was also highly variable over the 30-year period, was synchronous and in-phase with DOC, revealing the close link between solutes and DOC sourced from the local catchment. Flows from an upstream mesotrophic reservoir were a key driver of DOC concentration over long timescales and contribute to flushing of higher DOC
water.  Our novel analysis of a rare long-term observational record for a prairie reservoir underscores the relationship between DOC and flow source, with catchment source waters linked to poorer quality lake water. If the prairies experience higher precipitation that yields more catchment runoff under a changing climate, higher allochthonous inputs of DOC, as well as inputs of nutrients and other solutes will further degrade water quality in the lake, making drinking water treatment increasingly difficult. Importantly, management to control water level and flood risk can be in tension with managing flows for water
quality, due to the need to reduce flushing inflows to manage water level and downstream flooding.  Advancing environmental flow rules may help balance competing goals in this water insecure region, already subject to major climatic and water stressors.

**Data availability**

The streamflow data used for flow reconstruction, wavelet analyses, and GAM in the study are not available online, but may
be requested via client.service@wsask.ca. Chemical data used for wavelet analyses and GAM from the water treatment plant are not publicly available. Supporting data are available via annual reports at https://www.buffalopoundwtp.ca/publications/annual-report. Supporting data can be made available to researchers, subject to agreements which may include non-disclosure agreements. Details of how to access the data will be made available prior to publication (Blair Kardash, blairk@buffalopoundwtp.ca).

**Author contributions**

**AAPB:** Conceptualization, Methodology, Software, Formal Analysis, Data Curation, Writing – Original Draft, Writing – Review and Editing, Visualization; **HMB:** Conceptualization, Methodology, Investigation, Data Curation, Writing – Review and Editing, Supervision, Project Administration, Funding Acquisition; **CJW:** Conceptualization, Methodology, Investigation, Data Curation, Writing – Review and Editing, Supervision, Project Administration, Funding Acquisition; **AN:** Formal
Analysis, Data Curation, Writing – Review and Editing.

**Competing Interests**

Dr. Baulch has been funded by the Buffalo Pound Water Treatment Corporation who uses water from this system in drinking water supply. She has also been funded by the Saskatchewan Water Security Agency, who is responsible for flow management



and decisions, including agricultural drainage. Dr. Whitfield also engages with both partners on science needs and water
security issues in the prairies.

## Acknowledgements

Research funding for this work was provided by NSERC Discovery Grants to Helen M. Baulch and Colin J. Whitfield, and an
NSERC Graduate Scholarship – Master's to Anthony A. P. Baron. Funding was also provided through an academic-industry
partnership with the Buffalo Pound Water Treatment Plant supported by the Mitacs Accelerate program, along with ongoing
funding of the Buffalo Pound Water Treatment Plant for operations. Additional funding was provided through the Career-
Launcher Internship program, offered by Colleges and Institutes of Canada as part of the Government of Canada's Youth
Employment and Skills Strategy. Many thanks are owed to Blair Kardash, Laboratory Manager at the Buffalo Pound Water
Treatment Plant, for stewardship of treatment plant data, insights on the Buffalo Pound Lake ecosystem, and cooperation
throughout the project, as well as the Buffalo Pound Water Treatment Plant for chemical analyses. We also thank John-Mark
Davies from the Saskatchewan Water Security Agency (WSA) for sharing perspective and knowledge from many years of
working in this watershed, and the WSA for provision of streamflow data. We also wish to thank Jonathon Walter and
Lawrence Sheppard for guidance on wavelet analyses. Finally, we acknowledge members of the SaskWatChe and Bigfoot labs
for their support and feedback from the inception of this work.

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
