# Peer review of "Novel Statistical Analysis Illustrates Importance of Flow Source for Extreme Variation in Dissolved Organic Carbon in a Eutrophic Reservoir in the Great Plains"

_EGUsphere, 2024_

## Author Comment (AC1)

**Reviewer #2**

*Summary*

This manuscript evaluates a long time series of lake water quality to learn about controls of short- and long-term changes in DOC concentrations. This is a good fit for HESS and of interest to a broader water science community.

I like the methodological approach and find results convincing. I have, however, some major concerns: The introduction does not clearly lead to the objectives, the results need improvement on their presentations and the discussion often lacks the clear relation to the results. This needs improvement.

Moreover, consider this thought:

The model describes the dependencies of a set of water quality variables but is partly looking at a hen-egg problem. Are high DOC concentrations triggered by high TP concentrations within the lake or are they responding to the same water source coming into the lake? The GAM cannot clarify cause-effect relationships – so it can be used for system understanding but not for predictions. This can be made more clearer in the manuscript.

For more specific results see my comments below.

It is nice to learn that the reviewer sees the results of being convincing and the work as being of broad interest. We thank the reviewer for the encouraging feedback and useful questions raised. We will be explicit that the GAM is not intended to be used for making predictions.

*Specific comments*

Abstract

L9: Better be specific and name it "concentration" trends or "flux" trends, if this is the better fit.

Yes, we can be specific about the type of trends.

L14ff: While the title puts stress on the novelty of the statistical approach, the abstract is not doing this. Would it make sense to add this aspect here?

Yes, we can include this in the abstract.

Introduction

I am not fully convinced by the line of argumentation in the introduction that does not define the lack of knowledge that is addressed by the objectives. Similarly the methodological approach is not clearly motivated by the problem that is addressed here. Finally, the hypothesis is not grounded in the state-of-the-art knowledge described before.

Yes, the hypothesis was also flagged by reviewer 1, and we will modify this. We can also refine the introduction to better lead the reader to the objectives we address through this work.

L29: I am not sure what "across systems" means here? Across regions? Across water compartments?

We will clarify that we mean across studies of different watersheds.

L32f: If "across regions" is meant, Temnerud and Winterdahl are maybe not the best fit as a reference as they look at Sweden only.

We can clarify here. The first part of this sentence is building on the previous, while the references relate to the search for common drivers.

L35ff: For hydrological changes this would be a potential additional reference: 10.1002/2017GB005749, for Nitrogen deposition this one: 10.1111/gcb.13758

Thanks for these suggestions, we will work to include them.

L47: Is landscape complexity the best term here? I understand landscape complexity more as the complexity and heterogeneity of a given landscape and not of complexity across landscapes. This may be termed more clearly.

Yes, thanks for this suggestion, we can differentiate between complexity and diversity here.

L59ff: DOM (why not DOC?) as a "master variable" and "limnological behavior" deserves more explanation.

It is more appropriate to use DOM as a master variable as it captures DOC and a broader assemblage. We can expand to be more explicit about limnological behaviour.

Methods

L84: What is the dominant land use here and is that related to the nutrient rich soils? Are there people living in the catchment and what happens to their wastewaters?

We will expand description to talk about the importance of agriculture as the dominant land use. There are no substantive sewage sources into the local area.

L85ff: Does the non-contributing areas are totally disconnected or connected via a groundwater pathway?

Groundwater flows are generally not important at the catchment scale here. We can clarify.

L90ff: Is there any regulation of water level/ flow at the dam of Buffalo Pond or is everything managed upstream only? Additionally, Eyebrow Lake is not mentioned in the text. Does this lake play a role in this complex lake system?

We can speak to Eyebrow Lake, which not a true lake (and thus a misnomer). This feature is considered in the derivation of Q-BP flows as the change of storage here is reflected in the flows entering Buffalo Pound from the upstream. We can also expand description of lake level regulation.

L97f: This last sentence does not make sense for me when not underpinned with facts and referenced.

We can include DOC info to better support this.

L109ff: I miss information how samples where taken and how often. How many samples are averaged on the mentioned monthly base and how was averaging done?

We will add a sentence on the collection procedure.

Fig. 1: No need to write where the middle of the lake is when coordinates are given on the axis. Maybe mention the yellow intake point in the captions and define its abbreviation.

We will make these changes to the caption.

L138ff: I am a bit puzzled by the flow reconstruction. Why is BP inflow the reference point since the catchment contributing to the water in the lake seems to be larger (Fig. 1, areas contributing downstream of BP inflow). This needs further explanation. I note that in lines 166-174 there is a section on ungauged flows downstream of BP inflow but it is not explained how this QLC was used in the analysis. I note that this is part of chapter 2.4. … for the sake of understanding I suggest to first describe what is needed for the analysis and then to describe how this data is constructed.

We can provide additional detail here to clarify, and have identified above in response to reviewer 1 a need for a revised equation on Q-LC which will help in this regard. In effect, our analysis looks at inflows sourced from and upstream reservoir (Q-LD), local catchment inflows (Q-LC), and the major water source upstream of the lake (Q-BP), which represents a combination of those flows, but one that is dominated by Q-LD. In short BP is an important reference point because it represents gauged inflow nearest to the upstream inflow to the lake. It would be a shortcoming to not consider each of these independently in our analysis. We will describe the what and why of why each is considered in the analysis in the revised manuscript.

L184f: This text would profit from an explanation why GAM has been used. I note that this is described later but you also justify the use of wavelet analysis here – no reason not to do it for GAM as well.

Good suggestion, we will provide an explanation for our use of GAM here alongside that for wavelet analysis.

Results

L262: Consider a different header here. "Temporal parameters" sounds not too good for me.

We can change this to "Flow and water chemistry patterns"

Fig. 2: At this multi-annual scale it is hard to see the timing of the seasonal dynamics. Panels a and b are, to my understanding, managed flows to meet the water demand while panel c is a natural seasonal dynamic. Any idea how to show these differences? Maybe by a plot as day of the year in the SI? Consider to use the same y-scale for all discharge plots.

We can include a plot in the SI similar to the one used for Fig 3b to show within year variation.

L273: DOC concentration is described after discharge in the text but shown in Fig. 3 after showing all other constituents. Maybe show DOC (as the master variable here) earlier?

We feel it is important to describe flows at the start as this system is unique both due to region and management regime, so we are not keen on reversing Fig 2 and 3, but will ponder on this.

L300-301 and 303-304: Two sentences saying the same thing here. Maybe combine both.

We can revise the text to make more concise here.

L325ff: The text reads as if it is given that there is a clear driver-response relationship between predictors and DOC concentrations. However, for the constituent you partly look at a set of potentially connected variables. E.g. TP and Chl a can both describe algae biomass. NH4 may occur because algae break down.

DOC may be excreted by algae. TP may decrease when flow increases due to a dilution effect of wastewater sources… All these interactions mean that predictors are not independent and you partly look at a hen-egg problem. This is more part of the discussion but I suggest to spent effort in this text to avoid this clear driver-response style of writing.

We can revise the text to be more intentional in our description as predictors rather than drivers (at L335).

L339: What is the <~7 mg/L referring to? Root mean squared error?

No, this is a threshold where we noted a change in model performance, as in S4. We will include 'threshold' to be clearer in this regard.

Fig. 5: Consider to keep same colors for same constituents across the figures. Impressive fit of the observation by the way.

Good suggestion, we will update colour scheme here to correspond to that used for DOC in Figure 3.

Discussion

L355ff: This chapter 4.1 reads like a summary and conclusions. You make statements that are justified in later chapters. I find it more appealing for the reader to first argue and discuss and then make statements. From this text alone, the reader does not know the basis of your statements (which analysis the statements are referring to and how they were interpreted).

We can include in this text more explicit pointing to our results to better solidify the basis for the statements we offer in this section.

L369: Not a good sign when the discussion is the first time when the reader learns that there is agriculture in the catchment.

Yes, we note a previous question above about land use, which we will describe in the methods to be sure the reader better understands context around this system.

L384: Same is true for the lake residence time – this needs to come earlier.

Yes, we can describe residence time behaviour in our methods.

L392: This needs to be also part of the site description.

Yes, we can address this along with changes at line 97, about the nature of upstream flows that are diverted to Buffalo Pound Lake.

L394: That DOC can be allochthonous and autochthonous in lakes should be more explicitly part of the introduction.

We will expand the introduction to touch on the nature of DOC sources.

L395ff: Have you tried a DOC mass balance? This could strengthen the argumentation here? How high need concentration in the QLC be to be visible in the lake when most of the water is coming from LD?

Yes, this has been considered, and attempted to some extent, but we do not have robust flow data from the catchment, and residence times vary according to flows and ET losses, so this can not be done with satisfactory confidence. (there is limited monitoring data from the catchment, so that's a key gap).

L403ff: It would be helpful to learn about correlation among the predictors. You state that water inflow to the BP lake comes often from LD. So, how is QBP correlated with QLD and what are cumulative fractions of the lake water balance (eq. 1)?

We disagree about any value this would add (and refer to the rationale in our methods as to why coherence rather than correlation is the framework used). Our analysis highlights the importance of lags for the predictors, and these lags tell us that correlation is not the best approach to investigate these patterns. In the case of Q-LD and Q-BP specifically, the data in Figure 2a,b,c show quite clearly that Q-BP is dominated by Q-LD.

L427ff: This section reveals a large number of formerly undescribed data. I see that this data is not your own result but it may be helpful if there is an overview on average water quality from these unpublished sources as a table somewhere above and/ or the SI.

These data are now published, so we can update with citations.

L445ff: This section describes relationships between DOC and TP as source-driven mainly. Are there additional in-lake mechanisms as well? Joint release of DOC and TP from lake sediments under iron-reducing conditions? Or is this what you mean in L450f?

This is not what we mean at L450, but new work on TP release from sediments in the system has been done, and we can describe this as part of in-lake mechanisms.

L480ff: Brining in new data here in the late part of the discussion is not good. Your discussion should be based on everything you describe in the results plus literature.

This is not new data to the analysis, but rather data conducted through the broader research project. We will remove the supplementary figure here to help in this regard, and instead cite that work.

L475ff: The whole chapter 4.3 seems to be very detached from the results and discussion above. What of this information is directly related to your findings. I suggest to shift part of this to the problem definition in the introduction, to try to discuss how your results help with water management and omit the rest.

Thanks for this comment. We agree that the introduction should introduce the idea of flow management, as this would allow us to better tie in this section, and we will revise accordingly. Ultimately it is this section that is the 'why we care' of this work, as the DOC behaviour in the lake and how it is managed have tangible outcomes for humans relying on this water body for different purposes, so this section is key to closing that loop.

Conclusions

L519: This reads as if the upstream lake flushes DOC into the lake BP. Better use "diluting" here and name the source of the high DOC water.

Yes, we will revise the wording to improve clarity on the role of the upstream reservoir.

---

## Author Comment (AC2)

**Reviewer #1**

*Summary*

Title: Novel Statistical Analysis Illustrates Importance of Flow Source for Extreme Variation in Dissolved Organic Carbon in a Eutrophic Reservoir in the Great Plains, by Baron et al.

Baron et al. present long-term (1990-2019) chemical and hydrological data from the Buffalo Pound Lake, a drinking water source lake in the Canadian prairie region. By using novel statistical analyses, they aimed to find drivers of DOC concentrations in the lake at various temporal scales. Upstream regulated flow and several chemical parameters accounted for most of the variation in lake DOC concentration. They conclude that both flow regulation and natural processes in the face of a changing climate pose important challenges for drinking water treatability.

*Assessment*

This was an interesting read. Investigating drivers of DOC concentrations at short- and long-term scales is a recurrent but a relevant topic, more so in atypical areas such as the prairies. The manuscript is well-written, and appropriate and novel statistical methods have been used, which are well presented and justified. Yet, a few concerns and quite a few specific comments, which are probably not major overall, but would require some work before the manuscript can be accepted for publication. I therefore suggest the authors to consider my comments and amend the text accordingly or rebut.

Thanks for the encouraging feedback and ideas for improvements to the manuscript. We are glad you found this work interesting and easy to read.

**General comments**

The hypothesis that "changes in lake water chemistry would impact DOC at shorter timescales" is ambiguous. Changes in lake water chemistry can relate to processes happening in the catchment, which would be the ones driving both overall lake water chemistry and lake DOC concentrations (these processes would relate more to allochthonous sources of DOC). But changes in lake water chemistry can also relate to internal processes in the lake, which in turn can drive DOC concentrations (these processes would relate more to autochthonous sources of DOC). I would like to see more explicit hypothesis considering whether both or one of the group of processes are expected to be important. I would also like to see this differentiation more explicitly made throughout the discussion.

Thanks for this feedback. We will revise our hypothesis to "changes in lake TP would impact DOC at shorter timescales" as our hypothesis to explore a direct link between autochthony and DOC. We will better differentiate internal vs external drivers elsewhere.

In relation to that and as much as I would think it should be the case, the lack of relevance of the local catchment flow (Q-LC) to explain DOC concentrations in the lake appears to imply that in-lake processes are more important (?). The authors should reconcile this observation with the explanations they provide that argue that catchment processes drive DOC concentrations in the lake under certain conditions.

Our analysis indicates that it is the role of flow (Q-LD) that emerges as dominant at long timescales, while at short timescales DOC was in phase with or lagging behind water chemistry predictors. While Q-LC was not significant, the analysis tells us that Q-LD has a flushing/dilution effect. While DOC coherence with Q-LC did not emerge in the analysis, we can expand on potential explanations for this in the text, including that these flows are intermittent, and when there is flow the magnitude of these is not always such that it can drive a substantive shift in the chemical state of the lake. We can't infer from this that in-lake processes are more important than the role of Q-LC; the evidence suggests that both will contribute to elevated DOC concentrations in the lake. We can add text to enhance clarity on this in the revised manuscript.

In relation to that, I am left unconvinced of the mechanisms/processes/situations that relate to high DOC concentrations in the lake. Indeed, the statistical methods that the authors use generally fail at the upper range of DOC concentrations. On one hand, I would like to see a more explicit explanation of the circumstances that lead to higher DOC concentrations, and on the other hand, the authors should acknowledge at this upper range their analyses did not provide a satisfactory answer.

We have included in the supporting information details on GAM performance (Figure S3), and this is in agreement with the reviewer concern about performance at the edges of observed ranges in DOC. In the revised manuscript we will utilize this figure better, to draw the reader to the limitations at these levels, despite overall satisfactory performance. We can also expand the discussion to draw on recent work examining the behaviour of this system, e.g. with respect to in-lake processes.

The statistical methods were well-presented and justified. Yet, their results are difficult to follow at times. I would appreciate if the authors provide more analogies to how results would be presented when using more common methods. For example, how do predictors of DOC relate to DOC? Are they "positively" related, "negatively" related, something else? This is not clear in the text. Maybe an extra column specifying this in Table 2 would help?

We appreciate this question, and will be clearer in articulating that directionality of relationships is not available from this coherence analysis (as in correlation analyses). As such, this is not a detail that can be added.

In some parts, the connection of the discussion to the actual results was not fully clear. Please, consider making clearer links in this regard.

Yes, we will enhance the clarity here, and we will explore the suggestion that analogies can be applied to help provide clarity for the reader in this regard.

**Specific comments**

Title

Shouldn't it be "Novel Statistical Analysis Illustrates the Importance of Flow Source for Extreme Variation in Dissolved Organic Carbon in a Eutrophic Reservoir in the Great Plains". That is, please include "the" in front importance.

We will revise the title to include 'the'.

Abstract

L. 9. It would make sense to clarify that these trends have been overwhelmingly positive trends.

We can clarify that these trends have been mostly positive.

L. 10. They might not be universal, but I think there is little doubt that the prevailing driver was the decline in sulfur deposition and consequent increase in organic matter solubility.

We can be more explicit in our description of the importance of sulphate decline as a driver of increasing DOC trends via OM solubility, and that they have not been observed universally across regions.

1 Introduction

L. 25-38. When describing "the debate over the factors that govern DOC concentrations", one must consider that such drivers operate on varying temporal and spatial scales (see e.g. Clark et al., 2010, doi: 10.1016/j.scitotenv.2010.02.046). Thus, drivers are not necessarily exclusive, they might just be dominant at different temporal and spatial scales. Elaborating on my previous point, there is little doubt that, at the long-term scale, increasing DOC trends observed across vast regions in the Northern hemisphere affected by acid deposition were driven by, indeed, reductions in sulfur emissions. This is especially true in smaller, forest headwater catchments. Areas affected by varying chloride or nitrogen deposition (mentioned in the paragraph) would behave similarly as they would trigger the same chemical effect on organic matter solubility and I therefore would consider them as analogous, not differentiated, drivers. Areas less affected by acid deposition of any kind where other drivers might come into play might of course show other patterns.

Thanks for this comment. We will revise the text to highlight the dominant role of changing acid deposition on DOC concentration patterns observed in industrial regions. We will better contrast these situations, with those without a history of high levels of acidic deposition of industrial origin, such as the study system we investigate herein where changing DOC levels have not been linked to regional scale behaviour across many acid-sensitive water bodies.

L. 45-46. Here you use both, catchment (rather UK English) and watershed (rather American English). Just use one of the two here and throughout the manuscript (it appears that you mainly use catchment so use that at every instance).

We will revise to only use catchment throughout.

L. 47. Are you referring to DOC exports or to concentrations here? You already mentioned before that "DOC export is highly correlated with precipitation and annual runoff", which is true and rather uninteresting because it is self-evident given that export = runoff x concentration, and runoff generally varies across a much wider range of values than concentration does.

Yes, we refer to DOC exports, and can revise the text to characterize this as a dependence, rather than correlation, for improved clarity. We will revise the introduction following from reviewer 2 to better situate the uniqueness of the landscape, as a contrast to the systems where much of the DOC research (that documenting decadal scale trends) has been conducted. This will better highlight the uniqueness of this landscape, the importance of few water bodies in the context of water security, and our relatively nascent knowledge of controls on wide temporal fluctuations in DOC concentration, e.g. directly through DOC export from the landscape, and indirectly via nutrient runoff.

L. 51-53. This is a very important point that I was eager to see. Do you have a reference to back this up? My perception of this system is that most of the area is hydrologically non-effective, i.e. I find half to be a low estimate.

We will supplement this with citations (Godwin and Martin 1975: Calculation of gross and effective drainage areas for the Prairie Provinces; PFRA 2008: Prairie farm rehabilitation administration (PFRA) watershed project–- areas of non-contributing drainage) and additional detail. Yes, the majority is non-effective, and this can vary tremendously at local scales, which we will describe more thoroughly as improved context for those less familiar with the region.

L. 70-71. But changes in lake water chemistry are concomitant to changes in DOC and therefore not necessarily drivers of DOC in the lake, i.e. they also depend on hydrological connectivity with the landscape and upstream sources, on processes occurring in the catchment, etc. Or you mean that in-lake processes are important for driving DOC concentrations?

We will clarify that we mean DOC concentration here, and cite work by Baron (2022) to describe that in-lake processes can affect DOC concentrations by ~1 mg $L^{-1}$ as water transits through this system.

2 Methods

L. 78. This is just out of curiosity for my own understanding. Can the climate of a region that receives only about 300 mm of annual precipitation still be classified as "subhumid". I would consider that in the range of arid or semi-arid regions. But probably the evapotranspiration is very low too despite the warm summers?

Thank you for raising this question. It is more accurate to classify the local climate of the study site as semi-arid. We will differentiate this from the regional climate which is described as sub-humid to semi-arid climate.

L. 86-87. Interesting and important remark. However, I find the sentence oddly constructed ("contributes flow in 1:2 runoff years"?). Can you rephrase?

Yes, this can be revised to be clearer using a description that is accessible to non-hydrologists (e.g., "contributes flow in one out of two years on average over the long-term (median flow)").

L. 95-97. Can Lake Diefenbaker keep up with the demands from Buffalo Pound Lake under all circumstances?

Yes, the volume of Lake Diefenbaker is ~100 times greater than that of Buffalo Pound Lake, and receives continuous flow sourced from the Canadian Rockies. We will add this detail.

L. 100. In Figure 1, I assume Ridge Creek is a small tributary into the Qu'Appelle River (you also describe it in the text as such). It would therefore be helpful to represent it in the figure as a lotic water system the same as e.g. Iskwao Creek, i.e. with a blue line.

We have used the best available hydrometric layer available for the region, which unfortunately does not capture the (intermittent) flow path of Ridge Creek (or the entire flow path of Iskwao). We believe this is an important reminder of the complex hydrological regime of the system, and will provide additional description to the text so that the role of Ridge and Iskwao Creeks is not ambiguous.

L. 109-123. I assume water samples are filtered before they are chemically analysed. What is the pore size of the filter?

Pore size used is 0.7 micron. This detail will be added to the text.

L. 125-126. Required a complete record at what temporal scale? Monthly, as implicitly suggested? Please, specify.

We can specify that complete record here means 360 monthly averages of samples collected weekly.

L. 137-174. I very much appreciate the effort to get the hydrology right and the consideration of water mass balances and catchment (effective) contributing areas. There is just one thing I am not sure I understand. How come Q-BP (the inflow to the lake) that is very much influenced by Q-LD (the outflow from Lake Diefenbaker, which is outside the catchment area of BP) is included in equation 4 that attempts to estimate only the local catchment flows? Shouldn't Q-BP be Q-U (the ungauged portion) in this equation?

Thanks for catching this. Equation 4 has not been expressed quite accurately. The local catchment flow is estimated using a combination of ungauged flows from the local catchment (which was mistakenly expressed as a second Q-BP term) and gauged (Q-RC, Q-IC) flows from the local catchment but does not explicitly include Q-LD flows that contribute to the flow at QBP. We will revise this equation and supporting text. The equation will read as Q-LC ≈ Q-RC + Q-IC + Q-UC.

L. 176-179. Perhaps, remind the reader here that, for this analysis, you are using the monthly values that you estimated earlier.

Context will be added to note that this analysis is using monthly values.

3 Results

L. 265. Aren't both lakes covered with ice?

Yes, but flow continues through control structures during winter months. We can clarify this.

L. 263-272. Are typical peaks across the three Q generally associated with snowmelt events, or also with rainfall events?

Peaks at Q-BP and Q-LC are typically associated with snowmelt events, however heavy rainfall events do also contribute to peaks, particularly for Q-LC, albeit these are rare and typically smaller in magnitude than spring freshet. Detail to describing the timing of peak flows (e.g. top 5%) can be added to this paragraph or shown as a supporting figure.

L. 302-303. Are all these significantly coherent relationships found analogous to positive correlations or there are any negative correlations too? Is this something that can be said at all at this point? Either way, I think it is important to specify this for the reader.

Great question. Context will be added to indicate that, at this stage, positive or negative relationships can not be specified. Coherence relationships are analogous to the correlation strength (in terms of absolute value) but do not provide information on direction of the relationship. This is the primary reason GAM was used along with wavelet analysis. We will provide context to better link the wavelet and GAM results and clarify, where necessary, the limitations of each, and provide analogies to correlation strength and direction where appropriate.

4 Discussion

L. 359-360. How is climate having an overriding influence on DOC concentration? You have not analysed any climatic variable.

We are referring to prolonged periods of drought in the region that are often followed by years with heavy rains. We will be more precise in our description by articulating this, and that these climatic patterns influence flow management for Buffalo Pound Lake.

L. 355-388. This section makes an interesting description of the general context of BPL, but how does it relate to your results? I fail to see the connection.

Given the uniqueness of our region and study system, this context is an important precursor to the discussion that follows later. We can include in this text more explicit pointing to our results to better solidify the basis for the statements we offer in this section.

L. 390-405. Let me see if I understand this correctly. Q-LD would have a "negative" relationship with DOC concentrations at BPL, meaning that when it is the prevailing source of water to the lake (that is when the catchment is generally hydrologically disconnected), DOC concentrations are generally low. By contrast, when the catchment does hydrologically connect to the lake via the activation of e.g. Ridge Creek and Iskwao Creek, you would expect to have higher DOC concentrations from organic matter-rich catchment sources. However, you were not able to see this through your analyses. Is my interpretation correct? And if so, how is all of this reconciled?

Yes, your interpretation is correct. We reconcile this in that Q-LC does not emerge as a more dominant driver as this flow is transient, and much smaller in magnitude than that of Q-LD, so not all catchment flow events are large enough to drive meaningful change in the lake. One way to think of this is more limited power to detect the role of the catchment as a driver on the lake. We can expand our discussion to speak to these challenges and also include information demonstrating the relationship between Q-LC and Q-LD that further support conclusions about the importance of local catchment flow (When Q-LC is very high, Q-LD is decreased by water managers).

L. 406-412. I would appreciate here if you'd explicitly mention whether these synchronous or lagging patterns imply, in each specific case, that DOC and the corresponding chemical parameter both increase, both decrease, or they go in opposite directions at the time scales considered.

We will rewrite this text, to allow us to better link hypothesized mechanisms to the patterns (synchronous, lagging, positive, negative) by breaking down the individual areas for discussion and mechanisms - this will also involve some revisions to the text that follows (which we referred to as 'see below'), because we can now see it is challenging to follow.

L. 411-412. Still, local catchment flow was not a predictor of DOC concentrations in BPL.

Correct. We note that local catchment flows can be large but are often intermittent, thus signals from this water source are not clear. We can clarify that despite this, nutrient delivery from the local catchment still appears to be important because water from Q-LD is typically not nutrient-rich.

L. 415-416. Following a previous comment, it can be that DOC concentration in BPL is linked to other chemical constituents in the lake, but is it really driven by in-lake chemistry itself? On the other hand, you did not find a relationship with Q-LC. All to say that I am having difficulties reconciling all these results so I would appreciate if you can make it clearer.

Thank you for this comment. We can revise our use of drivers and instead use predictors.

L. 423-444. First, these are very high concentrations of sulfate compared to what I am used to in other natural environments. I assume this can only be explained by the geological settings of the region containing large amounts of gypsum and pyrite, i.e. the ultimate source of sulfate in the catchment and the lake should primarily be mineral weathering. If this is correct, please make it more explicitly clear in the text. Second, if I understood it right from section 3.3 and Figure 4d, sulfate has a complex relationship with DOC, where above certain threshold sulfate and DOC are "negatively" related and below this threshold they are "positively" related (excluding the upper sulfate concentration range where the model did not perform well). Is this correct and if so, how do you interpret it? I miss this explanation in this discussion.

Yes, this (1) is correct. We can add this detail to the discussion. In case 2, yes the relationship emerging in the GAM is complicated, and not one that is readily interpreted. Without stronger empirical evidence of any link between DOC and sulphate across the large range in sulphate concentrations presented here, we prefer to avoid speculation in any potential interpretation. There is likely to be very high spatial variability in sulfate in the catchment, which is contributing to this complex relationship.

L. 441-444. This might be the case, but how would this drive DOC concentrations in the lake? You need to provide support for in-lake control of DOC concentrations, if that's one of your lines of argumentation, which is still not clear to me.

We can expand this discussion to include the work of a recent student who explored salinity/sulphate dynamics across a range of pothole wetland types. The patterns at the scale of individual wetlands seems to suggest activation and delivery of sulphate to surface waters consistent with the temporal dynamics on record at Buffalo Pound Lake. We will also rephrase to describe as 'predictor' rather than 'driver'.

L. 449-450. But yet again, Q-LC was not related to DOC concentrations in BPL.

We can clarify here that nutrients delivered from local catchment are likely important despite intermittent catchment flows. From the collective comments from the reviewers, we also believe there is value in better explaining overall flow management, and that Q-LD is reduced with Q-LC is high. Adding these details will better prepare readers to navigate the analysis.

L. 451-473. Maybe these in-lake mechanisms are of greater importance than the catchment input mechanisms given the lack of relevance of Q-LC? I don't know, I am sceptical about that, but I am worried about the lack of explanatory power of Q-LC. In any case, you should be more explicit in differentiating catchment processes that can drive DOC concentration in the lake via allochthonous sources, and in-lake processes that can drive DOC concentrations via autochthonous sources. And once you make that differentiation clear, it would be best if you argue for either one of them with more conviction.

We have touched on this point (Q-LC not emerging as dominant) above, and this will be expanded on in the discussion. Likewise we can describe further the potential role of in-lake processes, we believe this will confirm the reviewers skepticism that that would be an outsized role for in-lake DOC production or processing.

L. 480. This was not entirely clear to me according to your results.

We can expand the text to link major water quality events with the wet-dry state and catchment flows.

L. 475-510. What I take from here is that this is a very challenging system in which no scenario is easy to manage. Would you provide a more explicit description of the conditions that would be best for both ecology and industry, even if they are not "natural" and potentially infrequent?

Yes, in line with the suggestions of shifting the introduction to better emphasize flow management, we will expand the discussion here, and can use some flow regime examples to describe potential trade-offs.

5 Conclusions

L. 5113-514. I would very much agree with this a priori, but given the lack of explanatory power of Q-LC, can you still claim that "pulses of allochthonous DOC from the local catchment during wet periods are linked to higher DOC"?

Through the responses above, we can do this in a number of ways. We can also strengthen our descriptions, as there are two prevailing flows. If we know that Q-LD acts to flush the system, then it stands to reason that (in the absence of inordinate amounts of DOC production in-lake), that these dilute upstream flows are countering the behaviour of local flows. We can be more explicit here in the revision.